

# Easy-to-use nomogram to predict neonatal hyperbilirubinemia

Shanshan Wang[1,*], Chan Wang[1,*], Siqi Zheng[2], Haiping Dou[1], Danyang Qu[1], Yuqian Wang[1] and Liu Yang[1]

[1] Department of Pediatrics, The Second Hospital of Dalian Medical University, Dalian, China
[2] Department of Otolaryngology, The Second Hospital of Dalian Medical University, Dalian, China
[*] These authors contributed equally to this work.

## ABSTRACT

**Background**. Neonatal hyperbilirubinemia is a common condition and a leading cause of hospitalization in newborns in their first week of life. Thus early identification of infants at risk is particularly important. In this study, we explored risk factors for its development of neonatal hyperbilirubinemia, and then constructed and validated an easy-to-use nomogram for the early prediction.

**Methods**. This study was conducted retrospectively and non-interventionally, involving 646 neonates born at the Second Hospital of Dalian Medical University between January 2021 and January 2024. The study population was systematically partitioned through cluster sampling into a training set comprising of 454 neonates and a validation set of 192 neonates, adhering to a 7:3 ratio, utilizing the R-4.4.0 program. Independent predictors of neonatal hyperbilirubinemia were identified using least absolute shrinkage and selection operator (LASSO) regression from the training set, and a nomogram was constructed based on these predictors. The performance of the nomogram was assessed using receiver operating characteristic (ROC) curves, calibration curves and decision curve analysis (DCA).

**Results**. Among 646 newborns, there were 350 males and 296 females, with a mean gestational age (GA) of 38.4 ± 1.4 weeks and birth weight (BW) of 3,264.1 ± 490.7 g. Six independent factors associated with hyperbilirubinemia were identified: GA, BW, premature rupture of membranes (PROM) ≥ 18 hours or concurrent maternal fever, maternal-infant blood type incompatibility with positive direct Coombs test, supplementation with probiotics, and weight loss > 9% within 3 days. Calibration curves indicated that the nomogram closely matched the actual observed values in both the training and validation sets. The areas under the ROC curves for predicting hyperbilirubinemia were 0.825 (95% confidence interval (CI) [0.777–0.874]) in the training set and 0.829 (95% CI [0.757–0.901]) in the validation set. DCA showed that the nomogram has clinical applicability.

**Conclusion**. The nomogram constructed in this study has good differentiation, calibration and clinical applicability, and has the potential to be used for predicting neonatal hyperbilirubinemia.

Corresponding authors
Yuqian Wang, 921269282@qq.com
Liu Yang, 439140628@qq.com

## INTRODUCTION

Neonatal jaundice is a common physiological phenomenon observed during the neonatal period, affecting almost all premature infants and roughly 60% of term newborns (*Olusanya, Osibanjo & Slusher, 2015*). Neonatal hyperbilirubinemia is defined as abnormally elevated levels of total serum or plasma bilirubin (TSB) and a leading cause of hospitalization within the first week of life (*Khan & Kim, 2022*). In recent years, the incidence of neonatal hyperbilirubinemia has been a rising (*Olusanya, Kaplan & Hansen, 2018*). If not identified and treated early, excessive free bilirubin can cross the blood–brain barrier, damaging neuronal cells, leading to kernicterus, resulting in complications such as hearing loss, vision impairment, or intellectual disability, which impose a significant burden on families and society (*Par, Hughes & De Rico, 2023*; *Kaplan, Bromiker & Hammerman, 2011*; *Qattea et al., 2022*). Therefore, accurate prediction and timely intervention are crucial.

Typically, the presence of hyperbilirubinemia can be initially assessed through visual inspection for jaundice of the newborn's skin, conjunctiva, or mucous membranes, as well as by monitoring the progression of jaundice from head to toe (*Par, Hughes & De Rico, 2023*). In recent years, advancements in medical technology and a deeper understanding of neonatal hyperbilirubinemia have led to the emergence of various methods that offer new possibilities for early diagnosis and intervention (*Du et al., 2021*; *Hulzebos et al., 2021*; *Kaplan et al., 2020*). Noninvasive detection devices (such as transcutaneous bilirubin (TcB) monitors) only assess bilirubin concentrations in the skin (*Okwundu et al., 2023*; *Dani, Hulzebos & Tiribelli, 2021*). At high TSB levels (greater than 250 µmol/L or 14.6 mg/dL), these devices have been shown to underestimate TSB concentrations, thus potentially affecting clinical decisions (*İşleyen et al., 2023*). Additionally, the accuracy of TcB measurements can be affected by skin pigmentation, lighting conditions, and/or user experience (*Khoshnoud Shariati et al., 2022*; *Ten Kate et al., 2023*). Traditional serological testing methods rely on biochemical techniques performed in laboratories, typically using enzyme-linked immunosorbent assays (ELISA) or colorimetric methods to quantify TSB levels. Although these approaches are currently the preferred methods for monitoring hyperbilirubinemia in neonates, they have some limitations. First, some measurement methods may yield inaccurate results due to operational errors, which can delay diagnosis (*Hegyi & Kleinfeld, 2022*). In addition, these testing methods are invasive as newborns are exposed to potential trauma and an increased the risk of infection due to blood collection. Finally, most current studies focus on using TcB levels to predict neonatal hyperbilirubinemia. Notably, it has been shown that combining TcB measurements and the presence of risk factors for neonatal bilirubin can significantly improve predictive accuracy. There are studies indicating that some risk factors, such as gestational age (GA), infection, and mode of delivery, may also have an impact on predicting neonatal hyperbilirubinemia (*Nickavar, Khosravi & Doaei, 2015*; *Han et al., 2015*). However, the specific risk factors that could be employed to anticipate the development of neonatal hyperbilirubinemia have not yet been identified. Therefore, there is a need to explore alternative noninvasive approaches.

In this study, we aimed to construct and validate an easy-to-use web-based nomogram using a number of risk factors to predict the risk of developing hyperbilirubinemia.

## METHODS

### Patients

The study included neonates born between January 2021 and January 2024 at the Second Hospital of Dalian Medical University with a GA $\geq$ 35 weeks. The diagnosis of hyperbilirubinemia was based on both clinical observations and serological evidence, following the recommendations of the 2022 American Academy of Pediatrics (AAP) clinical practice guideline revision (*Kemper et al., 2022*). Infants with congenital biliary atresia and those with severe malformations or chromosomal abnormalities were excluded. The study was conducted in accordance with the ethical principles of the Declaration of Helsinki and was authorized by the Ethics Committee of Second Hospital of Dalian Medical University (No. KY2024-325-01). The definitions of the stages of labor used in this study are as follows: the first stage of labor prolongation refers to more than 20 h for primiparous women and more than 14 h for multiparous women. The second stage of labor prolongation refers to more than 3 h for primiparous women and more than 2 h for multiparous women. Additionally, newborns were monitored for a duration of 2 weeks.

### Data acquisition

We performed a retrospective analysis of the medical records of all participating patients. The initial step involved contacting the guardian of each patient by phone to secure their consent for participation. Upon receiving agreement from the guardian, we sent the informed consent form for their consideration, signature, and subsequent return. A comprehensive literature review and analyses were conducted, and consultations took place with neonatal specialists using the case files of the infants to identify and summarize the risk factors linked to neonatal hyperbilirubinemia. Furthermore, we accessed the child's medical records and referred to the AAP clinical practice guideline of neonatal hyperbilirubinemia. This process led to the creation of the final version of our "Survey on Risk Factors for Neonatal Hyperbilirubinemia". Data collection was performed as previously outlined by *Zhang et al. (2023)*, which served as a research instrument to gather pertinent clinical information and establish a standardized data framework.

### Statistical analyses

The handling and analyses of data were conducted using R version 4.4.0, the developed by the R Foundation in Vienna, Austria, accessible at https://www.r-project.org/. Data were tested for normal distribution, and those that met normal distribution were expressed as mean $\pm$ standard deviation ($\bar{x} \pm$ SD) and compared between groups using the independent samples $t$-test, and those that did not meet normal distribution were expressed as median (M) as well as 25th and 75th percentiles (P25, P75). Categorical variables are articulated in terms of frequency and percentage (n (%)), employing the chi-square test, with the statistic denoted as the $\chi^2$ value. Continuous variables included GA, birth weight (BW), and maternal age. Categorical variables variables included fetal sex, umbilical cord blood gas

pH, Apgar score, blood group type with positive orthogonal resistance, cerebral hematoma, breastfeeding, probiotic supplementation, weight loss >9% within 3 days, cesarean section, gestational hypertension, gestational diabetes, amniotic fluid contamination, PROM ≥ 18 h or combined with intrapartum fever, placental abruption, placenta previa, prolonged labor, and umbilical cord abnormalities. Least absolute shrinkage with selection operator (LASSO) regression analysis was used to screen risk factors for neonatal hyperbilirubinemia. Random sequences were set and the study population was randomly divided into training and validation sets in the ratio of 7:3. Based on the training set data, the risk factors screened by LASSO regression analysis were subjected to multifactorial logistic regression analysis to determine the independent risk factors for neonatal hyperbilirubinemia, which in turn led to the construction of a clinical prediction model column-line diagram. Based on the training set and validation set data, the differentiation ability of the model was examined and calculating the area under the curves (AUCs), and the model was considered to have good accuracy if the AUC > 0.7; the calibration curves were plotted to evaluate the consistency between the actual results and those of the prediction model; decision curve analysis (DCA) assesses the clinical benefits of the model. $P < 0.05$ was considered a statistically significant difference.

The nomogram scores were derived from the logistic regression coefficients. Each predictor's score reflects its relative weight in the model, scaled to a 0–100 range for the smallest to largest effect. The total score maps to the predicted probability *via* a logistic function (*Iasonos et al., 2008*).

## RESULTS

### Clinical characteristics of the patients

This study included 646 newborns, and the consort diagram is illustrated in Fig. S1. The subjects were randomly divided into a training set (454 newborns) and a validation set (192 newborns) in a ratio of 7:3. The baseline characteristics are presented in Table 1. The mean GA of all newborns was 38.42 ± 1.38 weeks, and the mean BW was 3,264.09 ± 490.74 g. Based on the relevant diagnostic criteria for hyperbilirubinemia, newborns were categorized into two groups: those with and those without neonatal hyperbilirubinemia. A total of 164 (25.3%) newborns were diagnosed with neonatal hyperbilirubinemia, while 482 (74.7%) newborns were not.

### Comparison of clinical characteristics between newborns with and without hyperbilirubinemia

In the hyperbilirubinemia group, there were 92 males (56.1%) and 72 females (43.9%), with a mean GA of 37.54 ± 1.66 weeks and a mean BW of 3,049.85 ± 620.12 g. In contrast, the non-hyperbilirubinemia group was comprised of 258 males (53.53%) and 224 females (46.47%), with a mean GA of 38.72 ± 1.12 weeks and a mean BW of 3,336.98 ± 414.31 g. GA and BW were significantly different between the two groups ($p < 0.001$), while no significant difference was observed in sex distribution ($p > 0.05$) (Table S1).

In addition, significant differences were found in maternal-infant blood type incompatibility with direct Coombs test positivity (4.36 *vs.* 21.34%, $p < 0.001$), weight

**Table 1  Clinical characteristics of enrolled patients.**

| Factor | Training set (n = 454) | Validation set (n = 192) | t/χ² | P |
|---|---|---|---|---|
| GA (range, w) | 38.4 ± 1.4 (35.0–41.0) | 38.4 ± 1.4 (35.0–41.0) | −0.100 | 0.923 |
| BW (range, g) | 3,258.7 ± 503.5 (1,000.0–5,000.0) | 3,277.0 ± 460.2 (1,000.0–5,000.0) | 0.430 | 0.665 |
| Maternal-infant blood type incompatibility and Coomb's test (+), n (%) | 40 (8.81) | 16 (8.33) | 0.002 | 0.965 |
| Probiotics, n (%) | 244 (53.74) | 103 (53.65) | 0.001 | 0.992 |
| Weight loss >9%, n (%) | 8 (1.76) | 2 (1.04) | 0.108 | 0.742 |
| Cesarean section, n (%) | 130 (28.63) | 55 (28.65) | 0.003 | 0.989 |
| Meconium-stained amniotic fluid, n (%) | 45 (9.91) | 27 (14.06) | 1.947 | 0.163 |
| Gestational diabetes, n (%) | 60 (13.22) | 18 (9.38) | 1.531 | 0.216 |
| Gestational hypertension, n (%) | 242 (53.30) | 99 (51.56) | 0.102 | 0.750 |
| PROM ≥ 18 h OR maternal fever, n (%) | 63 (13.88) | 25 (13.02) | 0.027 | 0.870 |

Notes.

Abbreviations: GA, gestational age; BW, birth weight; W, weeks; g, gram.

loss exceeding 9% within the first three days (0.41 *vs.* 4.88%, $p < 0.001$), and probiotic supplementation (62.03 *vs.* 29.27%, $p < 0.001$). No notable differences were found between the two cohorts regarding umbilical blood pH < 7.2, Apgar scores ≤ 7, instances of cephalohematoma, and breastfeeding ($p > 0.05$) (Table S2).

## Comparison of maternal clinical characteristics between newborns with and without hyperbilirubinemia

A comparative analysis of maternal clinical characteristics between the hyperbilirubinemia and non-hyperbilirubinemia groups revealed statistically significant differences between rates of: cesarean section (25.31 *vs.* 38.41%), gestational hypertension (9.75 *vs.* 18.90%), gestational diabetes (55.39 *vs.* 45.12%), meconium-stained amniotic fluid (12.86 *vs.* 6.10%), or prolonged rupture of membranes (PROM) ≥ 18 h or concomitant intrapartum fever (9.13 *vs.* 26.83%) ($P < 0.05$). There were no statistically significant differences between the two groups regarding maternal age, gestational hypothyroidism, placental abruption, placenta previa, prolonged labor, and cord abnormalities ($p > 0.05$) (Table S3).

## Development of a nomogram for predicting neonatal hyperbilirubinemia

Six predictor variables with non-zero coefficients were identified using LASSO regression analysis with vertical curves plotted at the λ-minimum (λ = 0.046), including GA, BW, PROM ≥ 18 h or concurrent maternal fever, maternal-infant blood type incompatibility with positive direct Coombs test, supplementation with probiotics, and weight loss >9% within 3 days (Fig. 1). Based on the results of the LASSO regression analysis, the logistic regression equation (Table S4) was established using the "stepwise regression method",

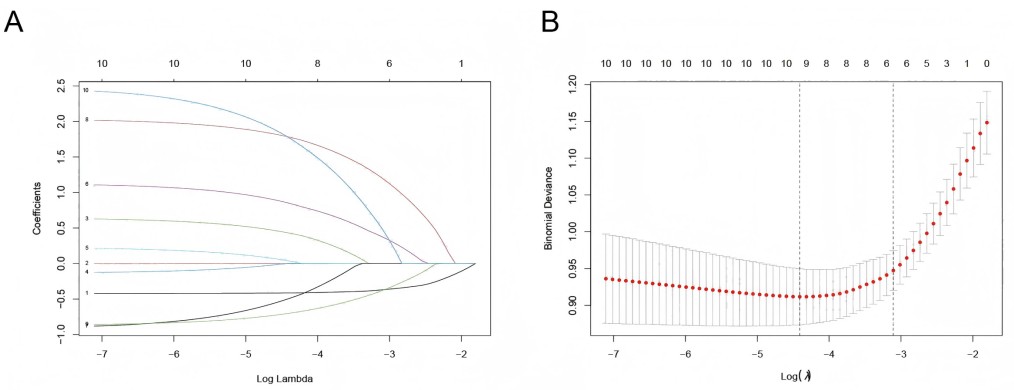

**Figure 1  Selection of predictors by the LASSO regression.** (A) LASSO coefficient profiles of clinical variables. (B) Identification of the optimal lambda in LASSO regression analysis using 10-fold cross-validation *via* minimum criteria. Note: 1, GA; 2, BW; 3, Hypertension in pregnancy; 4, Gestational diabetes; 5, Hypothyroidism in pregnancy; 6, PROM ≥ 18 h OR maternal fever maternal-infant blood type; 7, Amniotic fluid situation; 8, Maternal-infant blood type Incompatibility AND Coomb's test(+); 9, Probiotic supplementation; 10, Weight loss > 9% in 3 days.

and a user-friendly nomogram for neonatal hyperbilirubinemia was drawn to visualize the model, as shown in Fig. 2.

The nomogram assigns a value to each predictor variable based on the logistic regression model, and an example is shown in Table S5. In the training set ($n = 454$), the median score was 98 (IQR: 73–135) with a score range of 20–282, while in the validation set ($n = 192$), the median score was 99 (IQR: 74–125) with a score range of 27–267. Neonates who developed hyperbilirubinemia had significantly higher scores than those without hyperbilirubinemia.

## Evaluation and validation of the nomogram

The discriminatory power of the nomogram was assessed using receiver operating characteristic (ROC) curve construction. The training set had an area under the ROC curve (AUROC) of 0.825 (95% confidence interval (CI) [0.777–0.874]). The Youden index was calculated as sensitivity + specificity −1. The maximum value corresponds to a threshold probability of 0.262. This threshold yielded a sensitivity of 73.7% (95% CI [0.658–0.817]) and a specificity of 82.1% (95% CI [0.780–0.862]). Further, according to the column line graph model calculations, when the predicted probability was equal to this optimal cut-off value (0.262), the score corresponding to the total score scale of the column line graph was 118.34 points (Fig. 3A). Additionally, internal validation was performed using data from the validation set (Table 1), where the predictive model exhibited satisfactory performance, with an AUROC of 0.829 (95% CI [0.757–0.901]), sensitivity of 65.2% (95% CI [0.515–0.790]), and specificity of 90.4% (95% CI [0.856–0.952]), as shown in Fig. 3B. In both training and validation sets, the model demonstrated strong discriminatory ability.

The predictive model was evaluated using calibration curves (Fig. 4). The results indicated that the predictions made using the nomogram only slightly differed from the actual observed outcomes, demonstrating good consistency between the two datasets. Moreover, the Hosmer-Lemeshow goodness-of-fit test demonstrated that the model

A

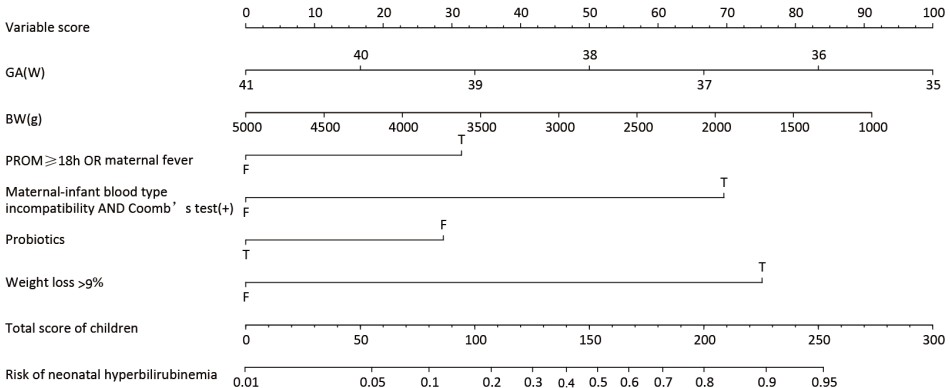

B

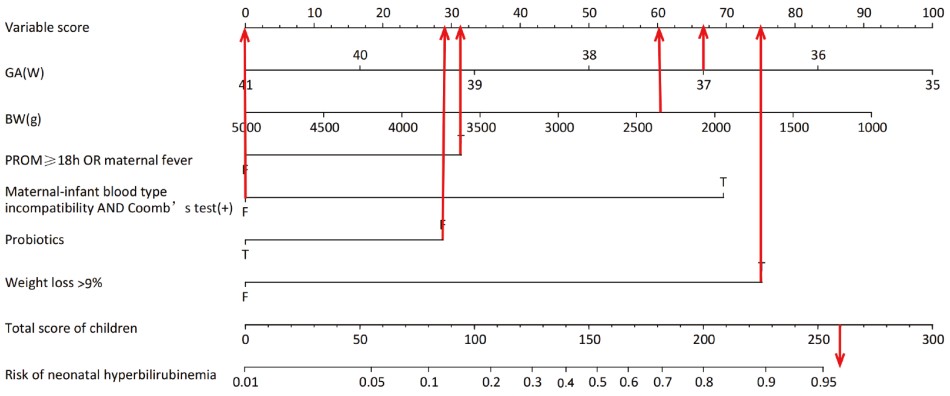

**Figure 2** **Nomogram for the prediction of neonatal hyperbilirubinemia.** (A) Nomogram for the prediction of neonatal hyperbilirubinemia. (B) A screenshot of the web-based nomogram. For example, a newborn at GA of 37+2 weeks with an BW of 2,380 g, had PROM for 22 h, no maternal maternal-infant blood type incompatibility, was exclusively breastfed after birth, was not given oral probiotics, and currently weighs 2,100 g. Red arrows show the scores for the indicators mentioned above. The total risk of hyperbilirubinemia in this child is about 260 points, and the risk of developing hyperbilirubinemia is greater than 95%. T = true; F = false.

exhibited a satisfactory fit in the training set ($P = 0.766$), while also showing good accuracy in the validation set ($P = 0.443$). The nomogram is shown in Fig. 5. DCA revealed that the threshold probabilities for predicting neonatal hyperbilirubinemia were 8%–78% in the training set and 7%–76% in the validation set.

## DISCUSSION

Currently, there is insufficient studies on evaluating risk prediction models for neonatal hyperbilirubinemia, both domestically and internationally (*Liu et al., 2022*; *Chou, 2020*).

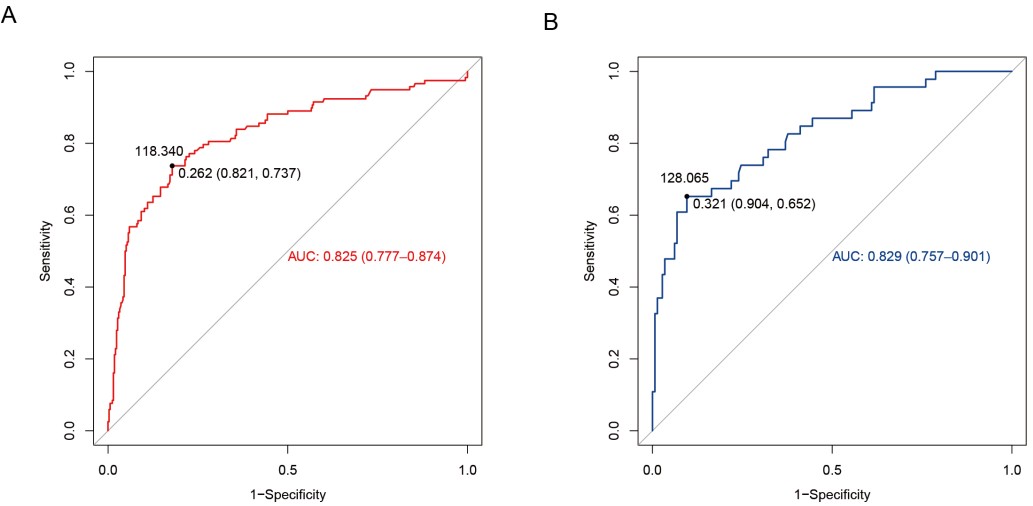

**Figure 3** Differentiation curve of the nomogram for predicting neonatal hyperbilirubinemia in the training (A) and validation (B) sets.

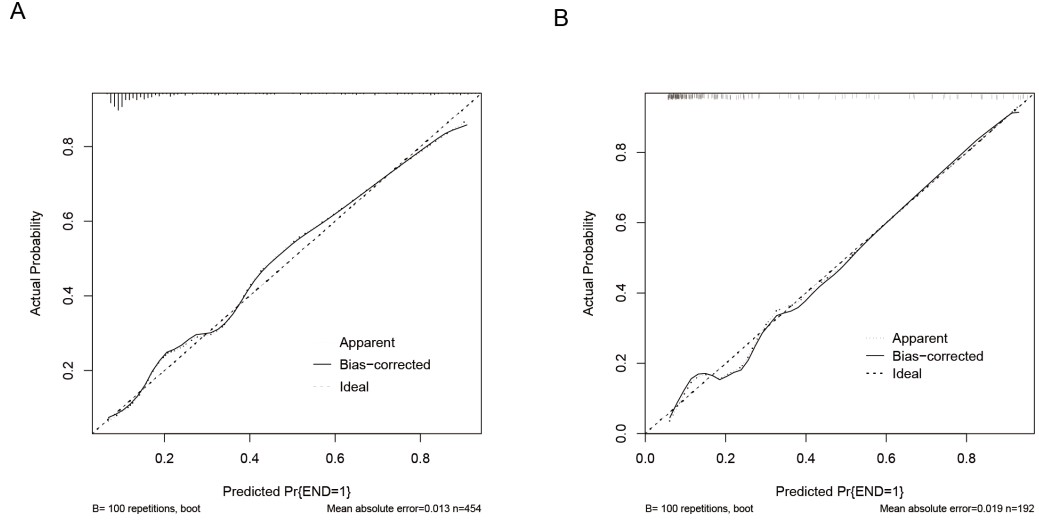

**Figure 4** Calibration curves of the nomogram for predicting neonatal hyperbilirubinemia in the training (A) and validation (B) sets. Note: the 'bias-corrected' calibration curve represents an adjustment made to the apparent (or naive) calibration curve to account for the overoptimism (bias) inherent when evaluating model calibration performance on the same data used to train the model.

Here, we used LASSO regression analysis to identify six independent predictive factors: GA, BW, PROM ≥ 18 h or concurrent intrapartum fever, maternal-infant blood group incompatibility with direct Coombs test positivity, weight loss > 9% within three days, and probiotic supplementation. Subsequently, we utilized these factors to construct a nomogram. We validated the risk prediction model using data from both the training and validation sets, assessing its discrimination, calibration, and clinical applicability. The

A

B

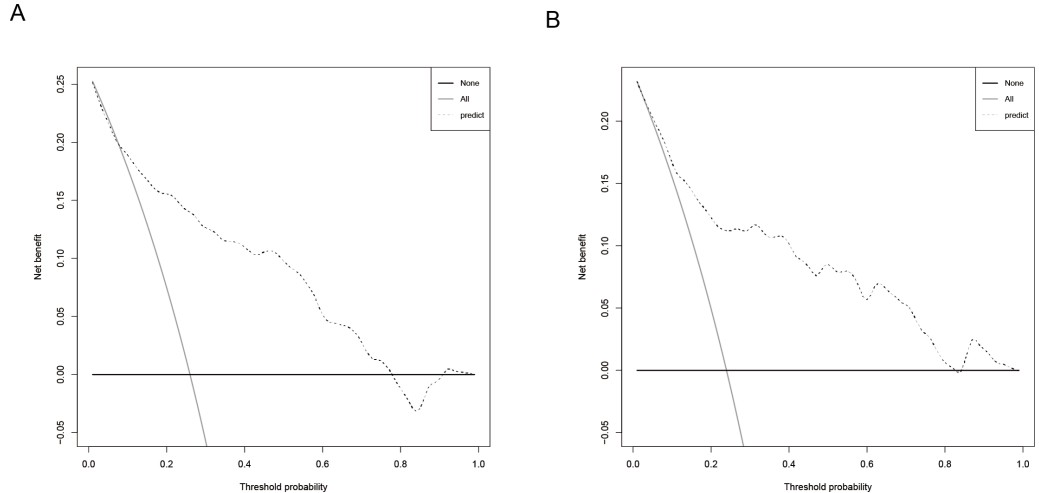

**Figure 5** Decision curve of the nomogram for predicting neonatal hyperbilirubinemia in the training (A) and validation (B) sets.

model was then evaluated and validated using ROC, calibration, and DCA. These findings are consistent with previous domestic and international studies (*Olusanya, Osibanjo & Slusher, 2015*; *Almohammadi et al., 2022*; *Boskabadi, Rakhshanizadeh & Zakerihamidi, 2020*; *Bergmann & Thorkelsson, 2020*; *Campbell Wagemann & Mena Nannig, 2019*). The model demonstrated strong performance in both sets, displaying excellent predictive capability.

Notably, the occurrence of neonatal hyperbilirubinemia was associated with maternal-infant blood group incompatibility, PROM, and neonatal infections, all of which have been shown in different studies (*Boskabadi, Rakhshanizadeh & Zakerihamidi, 2020*; *Bergmann & Thorkelsson, 2020*). In our study, we specifically focused on GA, BW, and PROM as predictive factors, which have also been identified as significant risk factors for hyperbilirubinemia in previous studies (*Ozdemirci, Kut & Salgur, 2016*; *Castillo et al., 2018*; *Yan, Deng & Hong, 2022*). Additionally, we found that probiotic supplementation may help protect neonates from hyperbilirubinemia. The gut microbiota plays a crucial role in bilirubin metabolism, and probiotics can influence gut motility and the gut microbiome, potentially preventing the onset of neonatal hyperbilirubinemia (*Zhang et al., 2022*). The mechanisms may include lowering intestinal pH, thus affecting β-glucuronidase activity, participating in bile acid metabolism, regulating the enterohepatic circulation, and enhancing the liver's ability to conjugate bilirubin (*Chen & Yuan, 2020*; *Santosa et al., 2022*; *Su et al., 2024*). Probiotics have recently gained popularity as a treatment for hyperbilirubinemia (*Suganthi & Das, 2016*; *Nasief et al., 2024*; *Armanian et al., 2016*). Our findings indicate that weight loss >9% within three days was an independent risk factor for hyperbilirubinemia, while BW acted as an independent protective factor. This may be related to the immature oral and swallowing coordination in newborns, along with their underdeveloped gastrointestinal systems, which can hinder adequate breastfeeding and result in varying degrees of dehydration and weight loss, subsequently increasing the

enterohepatic burden of bilirubin (*Gartner & Herschel, 2001*; *Chang et al., 2012*). This is consistent with existing literature, further enhancing the feasibility of our model.

Strengths of our study is that we combined several known risk factors to create a model and constructed a user-friendly web-based nomogram. The precision and discriminative ability suggest that the nomogram possesses extensive applicability and has potential to improve management of neonatal hyperbilirubinemia significantly. Within the training set, specificity was 81.5% with an AUROC of 0.822; conversely, with the validation set, specificity increased to 87.0%, with an AUROC of 0.840. For newborns with hyperbilirubinemia, the training and validation sets demonstrated enhanced predictive ability while maintaining precision. Furthermore, we also utilized a range of methodologies to assess the effectiveness of the nomogram. Calibration curves demonstrated a robust alignment between predicted and actual observed values, and DCA demonstrated high clinical applicability. In this nomogram, each predictive factor is a routine component used in clinical practice in this nomogram. This nomogram is not a replacement for the phototherapy thresholds recommended in the 2022 AAP guideline, but rather a tool for predicting whether a neonate is in the high-risk group and to guide clinicians in making decisions about pre-discharge risk or optimization of the timing of phototherapy initiation. For example, the probability of neonates developing hyperbilirubinemia after discharge was assessed by using the nomogram, and a risk of > 50% suggested the need for close monitoring of bilirubin levels (*Wickremasinghe et al., 2018*; *Novak et al., 2025*).

We must acknowledge certain limitations in our study. First, being a retrospective study, some patients with incomplete data may have been excluded, which could lead to bias. Therefore, caution should be exercised when interpreting our results. Second, there may be some other potential risk factors that were not included in our prediction model, such as sepsis, etc., but were not available for all patients. Lastly, we did not accurately assess the standardization of risk probability thresholds in relation to the timing of phototherapy initiation, and follow-up.

In conclusion, by utilizing readily available clinical parameters, we developed a nomogram that may serve as a valuable tool for predicting neonatal hyperbilirubinemia. Notably, this nomogram has been converted into a dynamic web-based tool, making it both convenient and easy for clinicians to use. However, among the six independent factors associated with hyperbilirubinemia, the direct Coombs test is more expensive and supplementation with probiotics may be not applicable in most lower-middle-income countries, thus limiting the global use of the nomogram. In conclusion, we established an easy-to-use, web-based nomogram with relatively good accuracy, which can help clinicians identify infants at risk for developing hyperbilirubinemia and thus improve their clinical outcomes.

### Funding

This work was supported by grants from the National Natural Science Foundation of China (grant number 82201896), the Foundation of Liaoning Province Education Administration

(grant number LJKMZ20221294). The funders had no role in study design, data collection and analysis, decision to publish, or preparation of the manuscript.

## Grant Disclosures

The following grant information was disclosed by the authors:
The National Natural Science Foundation of China: 82201896.
The Foundation of Liaoning Province Education Administration: LJKMZ20221294.

## Competing Interests

The authors declare there are no competing interests.

## Author Contributions

- Shanshan Wang performed the experiments, prepared figures and/or tables, and approved the final draft.
- Chan Wang performed the experiments, prepared figures and/or tables, and approved the final draft.
- Siqi Zheng analyzed the data, prepared figures and/or tables, and approved the final draft.
- Haiping Dou analyzed the data, prepared figures and/or tables, and approved the final draft.
- Danyang Qu analyzed the data, prepared figures and/or tables, and approved the final draft.
- Yuqian Wang conceived and designed the experiments, authored or reviewed drafts of the article, and approved the final draft.
- Liu Yang conceived and designed the experiments, authored or reviewed drafts of the article, and approved the final draft.

## Human Ethics

The following information was supplied relating to ethical approvals (*i.e.*, approving body and any reference numbers):

The studies involving human participants were reviewed and approved by The Ethics Committee of Second Hospital of Dalian Medical University (KY2024-325-01).

## Data Availability

Supplementary Files.

## Supplemental Information

Supplemental information for this article can be found online at http://dx.doi.org/10.7717/peerj.20017#supplemental-information.

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
