# Peer review of "Easy-to-use nomogram to predict neonatal hyperbilirubinemia"

_PeerJ, doi:10.7717/peerj.20017_

## Round 0.1 · original submission · Major Revisions

The present manuscript, which involves the development and validation of a predictive nomogram for neonatal hyperbilirubinemia based on a newly collected dataset, holds considerable clinical significance and has been positively recognized by the reviewers in that regard. However, there remain several areas that require further clarification and elaboration, and at this stage, we have determined that a major revision is necessary before the manuscript can be considered for acceptance.

Reviewer 1 ·

Basic reporting

This paper used a new dataset of newborn babies to develop a nomogram model for predicting neonatal hyperbilirubinemia. The authors firstly had the dataset analyzed and profiled clinical characteristics by comparing the clinical characteristics between newborns with or without the hyperbilirubinemia. Then, through detailed analysis and extracting key information, they developed a nomogram using LASSO regression analysis with six most predictive variables with non-coefficients from 10 variables. Finally, the paper validated using data from the validation set. Although this paper brought in a new set of data and established a nomogram using new variables, there are some issues about this manuscript

Experimental design

1. The paper claim this is a website-based tool but they did show how and where it can be done in details. It would be better if they add this part to the paper.
2. The sample size as the author mentioned is limited and not a lot of information was given on the validation data set that the paper mentioned on line158.
3. Another issue is that compared to previous published similar paper, the accuracy of the model this paper brought up seems to be lagging behind

Validity of the findings

Overall, this paper is well written and brought a new set of data and perspectives in predicating neonatal hyperbilirubinemia.

·

Basic reporting

General comments:
 It is a novel web-based nomogram with dynamic features, with rigorous internal validation (ROC, calibration, DCA) and focuses on available clinical variables (GA, BW, PROM).
 However, the reviewer suggests the concerns need to be addressed for the betterment of the document
 The document has no line numbers to make the communication between the author(s) and reviewer(s) easier.

Title: The reviewer suggests replacing advanced with severe or significant:
advanced seems ambiguous, and it does not fit with hyperbilirubinemia classification:

Experimental design

Introduction:
• The phrase “While these methods are sensitive and specific, making them the preferred option for bilirubin detection, they may cause errors and delays in practice” needs expansion and paraphrasing to make it clearer and convincing, as finding an alternative to the sensitive and specific diagnostic method seems weird for the reader.
• The reviewer requests the incorporation of more justification in the introduction section of the study (specifically, expand upon the knowledge gap or the improvements to be made by the current model compared to the existing models).

Methods:
Statistical analysis: The reviewer appreciates that the author utilised LASSO regression, which is suitable for developing a nomogram in scenarios with many predictors, enhancing the model's reliability. However:
• It is kindly requested that the author clarify which continuous variables were skewed versus normally distributed.
• Additionally, please specify which variables require parametric tests and which require non-parametric tests, as they appear to be mixed in these descriptions.
• Providing a sample size power calculation as it enhances the justification of the sample adequacy/inadequacy, as the probability of being small was discussed by the author.
Some additional clarifications to be provided by the author(s).
1. Exclusion criteria: Could the author explain why newborns born at <35 weeks of gestation were excluded from the study, considering they are among the most affected by hyperbilirubinemia?
2. Variable age at collection: Did the author gather the data on the postnatal age or describe anything about it on hyperbilirubinemia, especially those with clinical jaundice, given that its physiological and pathological characteristics can vary based on age?
3. Consent and chart review: The chart review aligns with the retrospective design of this study. However, it was mentioned that consent was obtained from the families. Could the author elaborate on the consent process and how the chart review was performed?
4. Facility details: Could the author provide more details about the facility? Specifically, is newborn screening applied (if so, for what), what is the typical duration of stay post-delivery for spontaneous vaginal deliveries, and how are newborns followed up? These service-related factors have an association with the hyperbilirubinemia, as per AAP.

Data acquisition
“A literature review and analysis were performed, and with the infants' case files, consultations were held with neonatal specialists to identify and summarise the risk factors associated with neonatal hyperbilirubinemia. This process culminated in the development of a final version of the "Survey on Risk Factors for Neonatal Hyperbilirubinemia." We utilised this self-developed survey as a research tool to collect relevant clinical data from newborns, establishing a standardised data framework for collection.
• Please provide the reference, or would it be possible to categorise predictor variables according to the AAP guidelines into major, minor, and protective factors? A standard textbook such as Nelson also utilises this approach, which will enhance the replicability and robustness of the study.
• There are also factors that need to be integrated into your risk factor analysis. For instance, diagnoses at admission (e.g., sepsis). If the data are not available, could the author please acknowledge them in the limitations section?

Validity of the findings

Abstract:
Method: Please add the retrospective designs, as it is significant information for the interpretation of the study findings.
The phrase “Newborns born to high-risk pregnant women” needs a summary of the criteria to define high-risk pregnancy
Result: The inclusion of a 95% CI for the AUC of the ROC curve for the results makes the result more comprehensive or complete. Please add it.


Results:
• Add the 95% CI for the sensitivity of 65.2% and the specificity of 87.0%.
• Please check the interpretation of the Hosmer-Lemeshow goodness-of-fit test (P = 0.018)
Discussion:
Conclusion:
The author put a conclusion as “This is especially important in resource-limited settings, where access to more advanced diagnostic tools may be restricted.”
Among six independent factors associated with hyperbilirubinemia, this study identified the direct Coombs test (expensive) and supplementation with probiotics (not applicable in most of the LMICS). So, it is logical that this study contributes to the unmet needs, such as invasiveness, while still a challenge in relation to the financial constraints (the combs test is more expensive than the liver function test) and relying on predictor variables that we may not find in the LMIC (probiotics).
It is appreciable that the author focused on clinically accessible variables (GA, BW, PROM).
• Hence, it seems advisable to rephrase the conclusion to show such an effort to link it more with LMICS. However, it's better to mention the challenge of strong connection in LMICS to utilise a web-based tool.

• The reviewer noticed that the author could compare model performance to existing prediction tools, e.g., the Bhutani nomogram, to contextualise performance, which would enhance its impact.

·

Basic reporting

see detailed comments

Experimental design

see detailed comments

Validity of the findings

see detailed comments

Additional comments

Easy-to-use nomogram to predict advanced neonatal hyperbilirubinemia

Wang S et al
Neonatal hyperbilirubinemia is a common condition and one of the leading causes of hospitalization in newborns in their first week of life. Thus, early identification of infants at risk is particularly important. Our aim was to develop and validate an easy-to-use nomogram for the early prediction of hyperbilirubinemia in newborns. Newborns born to high-risk pregnant women were enrolled and randomly divided into training and validation sets. Independent predictors of neonatal hyperbilirubinemia were identified using Least Absolute Shrinkage and Selection Operator (LASSO) regression from the training set, and a nomogram was constructed based on these predictors. The performance of the nomogram was evaluated using receiver operating characteristic (ROC) curves, calibration curves, and decision curve analysis (DCA). Among 646 newborns, there were 350 males and 296 females, with a mean gestational age were (38.42±1.38) weeks and an average birth weight were (3264.09±490.74) grams. Six independent factors associated with hyperbilirubinemia were identified: gestational age, birthweight, premature rupture of membranes >18 hours or concurrent maternal fever, maternal-infant blood type incompatibility with positive direct Coombs test, supplementation with probiotics, and weight loss >9% within 3 days. These factors were combined to construct the nomogram for predicting hyperbilirubinemia. The calibration curves showed that the nomogram closely matched the actual observed values in both the training and validation sets. The areas under the ROC curves for predicting hyperbilirubinemia were 0.822 in the training set and 0.840 in the validation set. DCA suggested that the nomogram holds clinical applicability. Therefore, the authors concluded that their newly developed, easy-to-use web-based nomogram model may provide a valuable and practical tool for predicting neonatal hyperbilirubinemia.
The manuscript is interesting, but there are some issues that need to be addressed. Also see my detailed comments below. This nomogram is only applicable to infants > 35 wks’ GA. Please state. In addition, the authors need to better explain why a 7:3 ratio was chosen. Furthermore, do the authors know what the causes of the hyperbilirubinemia were, such as presence of hemolytic disease? Most importantly, what were the 10 significant risk factors, how were they selected? Were they those listed in the AAP guideline?
Minor Comments:
Title:
line 1: delete “advanced”.
lines 5 and 7: delete “.” after “China”.
line 8: delete “.” after “authors”.

Abstract:
line 14: delete “in…period”.
line 15: change “one of the” to “a”; “causes” to “cause”; insert “Thus,” before “early”.
line 16: insert “of infants at risk” after “identification”.
line 17: insert “neonatal” after “of”; delete “in newborns”.
line 22: change “evaluated” to “assessed”.
line 24: delete second “,”.
line 25: change “were” to “of”; insert “(GA) after “age”; delete “(” and “)” x 2; “an average”; insert “(BW) after “weight”; change “were” to “of”.
line 26: delete “gestational age (” and “)”.
line 27: delete “birth weight (” and “)”.
line 30: change “the” to “a”.
line 32: change “area” to “areas”; “curve” to “curves”; “was” to “were”.
line 33: change “suggested” to “showed”.
line 37: change “non-invasive” to “noninvasive”.

Introduction:
line 39: insert “observed” before “during”.
line 41: change “refers to an” to “is defined as”; “level” to “levels”; insert “total serum or plasma” after “of”; insert “(TSB)” after “bilirubin”.
lines 41 to 42: change “in…becoming” to “and”.
line 44: change “shown a” to “been”; delete “trend”; describe how “free bilirubin” can form.
line 45: change “neural” to “neuronal”.
lines 45 to 46: delete “, and…damage,”.
line 46: change “and” to “or”.
line 48: delete “for…complications”.
line 50: insert “for jaundice” before “of”; change “and” to “or”; “observing” to “monitoring”.
line 51: insert “the” after “from”.
line 53: delete “modern”; change “offering” to “that offer”.
line 54: change “non-invasive” to “noninvasive”; delete “jaundice”; change “measurements” to “(TcB) monitors”.
line 55: change “through…measurements” to “in the skin”.
line 56: change “jaundice” to “hyperbilirubinemia”; delete “in newborns”; change “bilirubin” to “TSB”; insert “space” after “250” and “or 14.6 mg/dL” after “L”.
line 57: change “bilirubin” to “TSB”; insert “thus” before “potentially”.
lines 57 to 58: change “decision-making” to “decisions”.
line 58: change “non-invasive jaundice detection” to “TcB measurements”; “influenced” to “affected”.
line 59: change “the operator’s” to “user”.
line 62: change “bilirubin” to “TSB”; delete “in serum”.
line 63: change “bilirubin” to “TSB”; “detection” to “measurements”; not sure what ismeant by “cause errors and delays”.
line 64: insert “because” before “furthermore”; delete “exposing”; insert “are exposed” after “newborns”.
line 65: change “increasing the” to “increased”.
lines 65 to 66: change “non-invasive” to “noninvasive”.
line 67: change “intended” to “aimed”.
line 68: change “the occurrence of” to “risk of developing”.

Methods
line 72: insert “(GA)” after “age”; change “selected” to “recruited”.
line 74: change “edition” to “revision”; “((Neonatal…from”.
line 75: insert “clinical practice guideline” after “Pediatrics”; delete “))”.
line 79: delete “in this study”.
line 82: delete first “the”.
lines 83 to 85: please revise, very unclear description.
line 86: change “, We”? to “, which we”.
lines 86 to 87: delete “this…survey”.
line 87: delete “from newborns,”.
line 88: change “establishing” to “and to”; delete “for collection”.
lines 89 and 90: change “analysis” to “analyses”.
line 91: change “were” to “are”.
line 93: change “detected” to “identified”.
line 94: delete “set”.
line 95: change “set” to “sets”.
line 97: what is meant by “advanced”? authors mean “severe”? if so, define.
line 99: why say “dynamic”?; change “conducted” to “established”.
line 100: what is “NIT”? why need to assess “liver fibrosis stages”?
line 102: change “areas” to “area”.

Results
line 108: change “, with…process” to “and the consort diagram is”.
line 110: not sure what is meant by “7:3 ratio”.
line 111: change “gestational age” to “GA”; delete “(” and “)”; change “birth weight” to “BW”.
line 112: delete “(” and “)”.
line 113: delete “(NHB)”.
line 114: delete “(NNHB)”.
line 117: delete “Hyperbilirubinemia”.
lines 118 to 119: delete “Among….not”.
line 120: delete “gestational age (” and “)”; second “(“ and “)”; “birth weight (” and “)”; third “(“ and “)”.
line 121: delete “(” and “)”.
line 122: delete third “(” and “)”.
line 123: delete “(” and “)”.
line 124: change “gender” to “sex”.
lines 126 to 127: delete “of…group”.
line 128: delete first “%”.
line 129: delete first “%”.
line 131: what time Apgar”.
line 132: change “and” to “or”.
line 134: delete “Hyperbilirubinemia”.
line 136: delete first “group”; change “group” to “groups”.
line 137: delete “several areas”; first “%” and third “%”.
line 138: delete second “%” and fourth “%”.
line 140: delete first “%”.
line 142: change “and” to “or”.
line 144: match previous subheading format.
line 145: delete “ ‘ ”.
line 147: delete “gestational age (” and “)”; “birth weight (” and “)”; “premature…(“ and “)”.
line 150: change “represented as” to “converted to”.
line 151: change “screenshotted” to “as shown”.
line 152: match previous subheading format.
line 154: change “area….)” to “AUROC”; define CI”.
line 159: change “AUC” to “AUROC”.
line 162: define “resamples”.
line 166: change “presented” to “shown”.
line 167: delete “using this model”.
line 168: delete “who”.

Discussion
line 171: change “research” to “studies”.
line 174: change “independent” to “these”; “visualized through” to “and constructed”.
line 177: not sure what is meant by “7:3 ratio”.
line 178: delete “gestational”.
line 179: delete “age (” and “)”; “birth weight (“ and “)”; “prolonged…(“ and “)”.
lines 187 to 188: change “premature…membranes” to “PROM”.
line 189: change “gestational age” to “GA”; “birth weight” to “BW”.
line 191: change “research” to “studies”.
line 194: change “jaundice” to “hyperbilirubinemia”.
line 195: change “PH” to “pH”.
line 197: change “jaundice” to “hyperbilirubinemia”.
line 199: change “birth weight” to “BW”.
line 204: change “credibility” to “feasibility”.
line 210: change “set” to “sets”.
line 213: delete “decision…(” and “)”.
line 217: change “as” to “being”.

Figure Legends
Figure 1: line 260: delete “the”.
line 262: change “used” to “using”.
Please provide key for each color line.

Figure 2: line 264: delete first “Nomogram….hyperbilirubinemia”.
line 265: delete “A screenshot of”.
line 266: change “GW” to “GA of”; “, with” to “and”; insert “-“ after “maternal”.
line 267: change “did not take” to “was not given”.
lines 268 to 269: not sure what is meant here; please clarify; do the authors mean to state “The total risk of developing hyperbilirubinemia is approximately 260 points and therefore greater than 95%.”?
I do not understand how the nomogram works. Also is not (A) the same as (B)?

Figure 3: Which is (A) and (B)?

Figure 6: Where is the legend? Please provide.

References
Change titles to sentence case: refs #4,13,14,16,18,19,20,23,25,26,28,29,30.
Provide page number for refs #14,16,26,29.

Table 1:
line 5: change “w” to “wk”.
line 8: change “maternal” to “Maternal”.
line 11: change “Supplementing” to “Supplemental”.
line 19: insert “-” before “diabetes”.
line 21: insert “-” before “induced”.

---

## Round 0.2 · Minor Revisions

Please respond to the concerns pointed out by reviewer 3.

**PeerJ Staff Note**: Please ensure that all review, editorial, and staff comments are addressed in a response letter and that any edits or clarifications mentioned in the letter are also inserted into the revised manuscript where appropriate.

**Language Note**: The review process has identified that the English language must be improved. PeerJ can provide language editing services - please contact us at [email protected] for pricing (be sure to provide your manuscript number and title). Alternatively, you should make your own arrangements to improve the language quality and provide details in your response letter. – PeerJ Staff

Reviewer 1 ·

Basic reporting

The author has addressed all my points and the manuscript has improved.

Experimental design

The author has addressed all my points and the manuscript has improved.

Validity of the findings

The author has addressed all my points and the manuscript has improved.

Additional comments

The author has addressed all my points and the manuscript has improved.

·

Basic reporting

General comments
The author (s) have made visible improvements to their manuscript based on the feedback provided by the reviewers. The authors have addressed many of the concerns raised in the initial review; however, some minor issues rAddressed all the concernsemain that require further clarification or enhancement.

Experimental design

Addressed all the concerns

Validity of the findings

Addressed all the concerns

Additional comments

Addressed all the concerns

·

Basic reporting

An easy-to-use nomogram to predict neonatal hyperbilirubinemia: A retrospective study

Wang S, et al
In this revision, the authors attempted to address all the concerns of the 3 Reviewers. As a result, the manuscript has not been improved, and almost in a poorer quality. I understand that it may be difficult to incorporate the concerns of all Reviewers, but a careful read should be done prior to submission. Their attention to detail is lacking and very concerning. The majority of the paper only outlines how the nomogram was developed, but no discussion is provided as to how I can apply this “easy-to-use” nomogram. For example, how is the “score” determined? Once I get the “score”, what do I do with it? Is it used as an adjunct to the 2022 AAP Clinical Practice Guideline to determine when to start phototherapy? Nothing is mentioned regarding the “management” options once I use the nomogram. Again, I recommend someone proficient in the English language proofread prior to submission as there are numerous typos and grammar issues throughout. See my detailed comments below.
Minor Comments:
Title:
Delete “A retrospective study”… that is obvious.
Abstract:
line 17: delete first “the”.
line 18: change “hyperbilirubinaemia” to “hyperbilirubinemia”.
lines 18 to 19: what is “predictive model risk column chart model”?
line 19: change “individualised” to “individualized”.
line 23: delete “, who…analysis”.
line 26: change “Least” to “least”.
line 27: change “Absolute… Operator” to lower case.
line 29: change “Receiver… Characteristic” to lower case.
line 30: change “Decision… Analysis” to lower case.
line 31: change “of” to “were”.
line 32: change “were” to “od”.
line 44: change “hyperbilirubinaemia” to “hyperbilirubinemia”.

Introduction:
line 57: change “sclera” to “conjunctiva”.
line 58: delete second “the”.
line 60: delete “diagnostic”.
line 62: change “concentration” to “concentrations”; delete “to…hyperbilirubinemia”3
line 63: delete “space”.
line 64: delete “the”; change “concentration” to “concentrations”.
line 65: delete “factors such as”.
line 66: change “and” to “and/or”.
line 67: delete “analysis”.
line 68: change “assay” to “assays”.
line 69: delete “symptoms of”.
line 70: What “errors”? What “delays”?
line 72: insert “due to blood collection” after “infection”.
line 73: change “transcutaneous bilirubin” to “TcB levels”.
line 74: change “transcutaneous bilirubin” to “TcB”; “with” to “the presence of”.
line 76: insert “(GA)” after “age”.
line 78: insert “the development of after “anticipate”; change “are” to “have”; insert “been” after “yet”.
line 80: change “network” to “web-based”; “based on” to “using”.
line 81: change “hyperbilirubinaemia” to “hyperbilirubinemia”.

Methods
line 84: change “subjects were” to “included”.
line 85: delete “gestational age (” and “)”.
line 86: change “identification” to “diagnosis”; “derived from” to “based on”.
lines 87 to 88: please provide correct title of the guideline.
line 90: delete “from this study”.
line 92: delete second “The”.
line 93: change “included” to “used”.
lines 99 to 100: delete “Given…study,”.
line 100: change “involves” to “involved”.
line 102: change “send” to “sent”; ; delete “for collection”.
line 103: delete “for our records”.
line 106: insert “proper” before “management”.
line 107: change “edition” to “version”; third “the” to “our”.
line 108: change “executed” to “performed”.
line 109: change “in” to “by”; what is “(DATE)”?
lines 111 and 112: change “analysis” to “analyses”.
line 113: delete “Measurement”.
line 114: change “data” to “those”.
line 115: what is “standard”? do you mean “standard deviation”?
line 116: change “data” to “those”.
line 117: change “percentile” to “percentiles”.
lines 118 to 119: please revise, very unclear.
line 119: delete fourth “The”.
line 120: insert “(GA)” after “gestational age”; “(BW)” after “birth weight” to “BW”; delete “The”.
line 122: change “incompatibility” to “type”.
line 127: change “hyperbilirubinaemia” to “hyperbilirubinemia”; what is meant by “seeds”?
line 131: change “hyperbilirubinaemia” to “hyperbilirubinemia”.
line 132: insert “ability” after “differentiation”.
line 134: insert “(AUCs)” after “curves”.
line 136: change “Decision” to “decision”.

Results
lines 147 to 148: delete “diagnosed…hyperbilirubinemia”.
line 149: insert “and without” after “with”.
lines 149 to 150: delete “and…Hyperbilirubinemia”.
line 153: change “comprised” to “was comprised of”; add parentheses around “53.53%” and “46.47%)”
lines 157 to 158: delete “several….factors:”.
line 160: change “or” to “and”.
lines 160 to 161: delete “Nevertheless….revealed”.
line 161: change “distinctions” to “differences were found”.
line 163: insert “and without” after “with”.
line 164: delete “and…Hyperbilirubinemia”.
line 165: insert “hyperbilirubinemia” after first “the”; delete second “the”.
line 166: insert “between rates of” after “differences”; delete “rates”.
line 174: change “screened out” to “identified”.
lines 174 to 175: delete “based…predictors”.
line 179: change “NHB” to “neonatal hyperbilirubinemia”.
line 180: insert “and” before “illustrated”.
line 181: insert “the” after “of”; delete “prediction… hyperbilirubinemia”.
line 182: change “nomograms” to “nomogram”; “analysis” to “curve construction”.
lines 182 to 183: change “The…demonstrated” to “The training set had”.
line 183: define “CI”; delete “in…set”.
line 184: delete second “Youden index”.
line 186: change “discrimination” to “discriminatory ability”.
line 192: what is meant by “500 bootstrap”?; insert “only” after “nomogram”.
line 196: delete “decision (” and “)”; change “. The DCA” to “, and”.
lines 198 to 199: what is meant by “Patients used”?

Discussion
line 201: insert “evaluating” after “on”.
line 203: delete “as…variables,”; delete “and”.
line 205: delete first and second “curves”.
line 206: delete “This…7:3”.
line 215: change “is” to “was”; delete “with…as”.
line 216: change “referenced” to “shown”.
line 217: change “research” to “study”; “particularly” to “specifically”.
line 219: change “our study” to “we”.
line 223: insert “thus” after “pH,”.
line 226: change “is” to “was”.
line 227: change “acts” to “acted”.
line 233: change “Our….First,.” to “Strengths of our study is that”.
line 234: what is meant by “readily available”?; change “presenting it as” to “constructed”
line 235: delete “outstanding”; “illustrated…curve”.
line 236: change “is poised” to “has potential”.
line 237: change “efficiency significantly” to “management of neonatal hyperbilirubinemia”; delete “recorded at”.
lines 237 to 238: change “, accompanied by” to “with”.
line 238: change “in” to “within”; “improved” to “increased”.
line 239: insert a space between “0.840” and “For”.
line 240: insert “also” after “we”.
line 243: insert a space between “DCA” and “demonstrated”.
lines 243 to 244: not sure what is meant by “each…testing”.
line 245: delete “model”.
lines 247 to 249: “We….data.”; this goes without saying, please delete.
lines 249 to 250: “However….study.”; please expand on why this is a limitation.
lines 250 to 252: “Second….research.”; please expand on why this is a limitation.
line 253: what is meant by “data are inconvenient”?
lines 254 to 257: “Lastly….hyperbilirubinemia.”; please expand on why this is a limitation.
lines 257 to 258: delete “Despite….developed.” limitations do not stop you from developing the nomogram. You developed a nomogram, but there are limitations of your study that may affect its usability.
line 261: change “serves” to “may serve”.
line 263: change “accessed as” to “developed into”.
line 268: insert a space between “countries.” and “In”; change “this study” to “we”.
line 269: change “novel…model” to “an easy-to-use, web-based nomogram”.
line 270: delete “to”; change “hyperbilirubinaemia at an early stage” to “hyperbilirubinemia”; insert “thus” after “and”; where in your study allows you to say “at an early stage”?

Author contributions
line 281: change “did Laboratory” to “performed laboratory”.

References
Change titles to sentence case: refs #4,13,14,16,18,21,22,23,25,26,28,29,31,32, and 33.
Provide page number for refs #14,18,29,32.
Also please follow page number formatting. All inconsistent throughout.


Figure Legends
Figure 1: line 407: delete “the”.
line 409: delete “used”.
Please provide key for each color line shown in Panel A

Figure 2: line 410: change “nomogram” to “Nomogram” x 2.
line 412: delete “GW“, “,”, change “and” to “an”; insert space between “2380” and “g”.
line 414: delete “did not take”.
lines 414 to 416: the authors need to describe better how 260 points are totaled.

Figure 3: line 418: delete first “set”; move “(B) to after ‘validation”; change “set” to “sets”.

Figure 4: line 420: delete first “set”; move “(B) to after ‘validation”; change “set” to “sets”.
What is meant by “bias-corrected”?

Figure 5: line 422: delete first “set”; move “(B) to after ‘validation”; change “set” to “sets”.
Inset on figure is cut-off, please fix.

Table 1:
line 1: change “included” to “enrolled”.
Please provide ranges of values for GA and BW.

Experimental design

See above

Validity of the findings

See above

Additional comments

See above

---

## Round 0.3 · Major Revisions

The manuscript still needs improvement as the reviewer suggested.

·

Basic reporting

An easy-to-use nomogram to predict neonatal hyperbilirubinemia

Wang S, et al
In this revision, the authors have attempted to address most of the concerns of this Reviewer and the manuscript is improved, but their lack of attention to detail is of concern and frustrating. I still recommend that the determination of “score” be better described in the text. What were the ‘scores” of the 2 cohorts? Please provide. I cannot recommend acceptance of this manuscript in its present form. I have detailed my concerns below. Again, I strongly recommend someone proficient in the English language proofread prior to submission as there are still numerous typos and grammar issues throughout. See my detailed comments below.
Minor Comments:
Abstract:
line 16: insert space between “for” and “the”.
line 17: change “to develop” to “constructed”; “validate” to “validated”.
line 19: change “investigation” to “study”.
line 22: insert “of” before “454”.
line 23: insert “the” before “R-4.4.0”; delete “for this process”.
line 29: change values to 0.1 decimal places.
lines 33 to 34: delete “These…hyperbilirubinemia.” redundant.
line 34: delete “The”.
line 36: define “CI”.
line 37: change “holds” to “has”.
line 40: insert space between “:” and “The”; change “clinical…model” to “nomogram”.
line 41: change “provides a reference” to “has the potential to be used”.

Introduction:
line 58: delete “,”.
line 59: change “primarily” to “only”.
line 60: delete “However,”.
line 61: change “may” to “have been shown to”.
line 67: change “Firstly” to “First”; “tools” to “methods”.
line 68: what “handling issues”?
line 69: change “;” to “as”.
line 71: change “transcutaneous bilirubin” to “TcB”.
line 72: insert “and” before “the”.
line 77: delete “for…hyperbilirubinemia”.
line 78: change “build” to “construct”.
line 79: change “hyperbilirubinaemia” to “hyperbilirubinemia”.

Methods
line 84: change “aligning with” to “following the recommendations of”.
lines 84 to 85: delete “update…:”.
line 87: change “Paediatrics” to “Pediatrics”; delete “professional standards”.
line 99: delete “will”.
line 101: change “analysis” to “analyses”.
line 103: change “guidelines” to “guideline”.
line 108: change “analysis” to “analyses”.
line 112: change “s” to “SD”.
line 129: what is “work characteristics”?

Results
line 135: delete “of the Patients”.
line 140: delete second “the”.
line 141: insert “and those without” after “with”; delete “and…hyperbilirubinemia”.
line 142: delete “Among…population,”.
line 144: change “with and without” to “With and Without”.
lines 149 to 150: delete “Statistical…in”.
line 150: insert “were significantly different” after BW”.
lines 150, 164, 167: change “P” to “P”.
line 152: change “When…characteristics” to “In addition”.
line 158: change “with and without” to “With and Without”.
line 168: change “nomogram for predicting neonatal hyperbilirubinemia” to “Nomogram for Predicting Neonatal Hyperbilirubinemia”.
line 174: delete “,”.
line 175: change “validation of the nomogram” to “Validation of the Nomogram”.
line 176: define “ROC”.
line 192: change “P” to “P” X 2.

Discussion
line 199: change “In…study” to “Here”.
line 200: change “P” to “P”.
line 201: delete “,”.
line 202: change “decision” to “DCA”.
line 232: change “wieth” to “with”.
line 237: change “In…practice” to “In this nomogram”.
line 238: insert “used in clinical practice” after “component”; change “This….graph” to “The nomogram”.
line 239: change “an advance” to “a tool”.
line 240: change “prediction…of” to “for predicting”; “belongs to a” to “is in the”; “in…help” to “and to guide”; “make better” to “in making”.
line 241: delete “screening”; change “and” to “or”; “optimisation” to “optimization”; insert “initiation” after “phototherapy”.
lines 241 to 244: please clarify what is meant by: “For example…levels”. Is this by using the nomogram? Or examples from refs 38 and 39?
line 242: change “hyperbilirubinaemia” to “hyperbilirubinemia”.
lines 245 to 246: delete “Although…hyperbilirubinemia”.
line 246: insert “in our study” after “limitations”.
line 247: please explain why “caution should be exercised”; delete one of 2 “,” after “Secondly”.
line 248: change “could…study” to “were not included in our prediction model”.
lines 248 to 249: change “are…are” to “were”; delete “, which…study.”
line 250: change “This….and” to “we”.
line 251: change “standardisation” to “standardization”.
line 252: delete “the timing of”.
lines 252 to 254: delete “Therefore,…charts”.
line 255: insert “by” before “utilizing”; change “the” to “we developed a”; insert “that” after “nomogram”.
line 256: change “innovative model can be” to “nomogram”.
line 257: change “developed” to “has been converted”.
line 258: delete “in their practice”.
line 259: delete “this…identified”; “(….test” and “)”.
line 260: change “(which is” to “may be”; delete “)”.
lines 261 to 262: change “. This…countries” to “thus limiting the global use of the nomogram.”.
line 264: change “neonatal” to “infants at risk for developing”; insert “their” after “improve”.

References
Please provide full page numbers for refs #5. 16, 17, 26, 35, 36, 37.

Figure Legends
Figure 1: I only see 9 variable lines on the graph, while legend provides 10.

Figure 2:
line 404: change “nomogram” to “Nomogram” x 2.

Table 1:
Please provide ranges of values for GA and BW in the same significant figure format as mean ± SD. Show only to 0.1 decimal place.

lines 2 to 3: Why “NHB”, “NNHB” shown? None of these abbreviations are shown on the Table.
line 3: delete “.”

Experimental design

See above

Validity of the findings

See above

Additional comments

See above

---

## Round 0.4 · Minor Revisions

Minor issues need to be addressed.

·

Basic reporting

An easy-to-use nomogram to predict neonatal hyperbilirubinemia

Wang S, et al
In this revision, the manuscript is improved, but there is still a lack of attention to detail as not all minor issues addressed. Please see below. There are still numerous typos and grammar issues throughout.
Minor Comments:
Abstract:
line 17: change “the” to “its”.
line 18: delete “of…hyperbilirubinemia”.
line 38: change “clinical…model” to “nomogram”.

Introduction:
line 68: change “error” to “errors”.

Methods
line 83: change “of…:hyperbilirubinemia:” to “American Academy of Pediatrics (AAP)”.
lines 83 to 85: delete “: management…)”.
line 87: change “authorised” to “authorized”.
line 101: insert “clinical practice” after “AAP”.
line 102: what is meant by “proper”?

Results
line 149: delete “were significantly different” after BW”.
line 150: insert “were significantly different” after BW”.
lines 164 and 167: change “P” to “P”.
line 176: change “score” to “value”; delete “its….to”.
line 177: change “the…principle” to “an example”.
line 178: delete “total nomogram” and “points”.
line 179: delete “points”.
line 180: please provide “scores” for both cohorts.
line 195: change “by” to “using”.
line 198: change “and” to “.”.
line 199: change “the” to “The”; delete “The”.
lines 201 to 202: delete “Patients….strategies”. You do not know this for a fact.

Discussion
lines 204 to 216: Paragraphs are somewhat duplicative. Please combine.
line 234: change “available” to “known”.
line 245: change “of” to “recommended in”.
line 248: change “hyperbilirubinaemia” to “hyperbilirubinemia”.
line 249: change “suggests” to “suggested”; “bilirubin” to “TSB”.
line 251: change “Being” to “being”.
line 252: change “be” to “have been”.
line 253: change “Secondly” to “Second”; insert “other” after “some”.
lines 254 to 255: delete “these…they”; insert “for all patients” after “available”.

Experimental design

see above

Validity of the findings

see above

Additional comments

see above

---

## Round 0.5 · accepted · Accept

After careful consideration, we are pleased to inform you that your manuscript has been accepted for publication. While the scientific content is now suitable for publication, we recommend that the text undergo minor language editing to address residual typographical and grammatical issues before final production.